# Analysis of the Promoter Regions of gga-miR-31 and Its Regulation by RA and C-jun in Chicken

**DOI:** 10.3390/ijms241512516

**Published:** 2023-08-07

**Authors:** Yingjie Wang, Ruihong Kong, Ke Xie, Cai Hu, Zongyi Zhao, Yuhui Wu, Qisheng Zuo, Bichun Li, Yani Zhang

**Affiliations:** 1College of Biotechnology, Jiangsu University of Science and Technology, Zhenjiang 212018, China; maggieyingjie@foxmail.com (Y.W.);; 2Jiangsu Province Key Laboratory of Animal Breeding and Molecular Design, College of Animal Science and Technology, Yangzhou University, Yangzhou 225009, China

**Keywords:** gga-miR-31, C-jun, RA, promoter, chicken

## Abstract

The role of gga-miR-31 in chicken germ cell differentiation and spermatogenesis is of significant importance. The transcriptional properties of gga-miR-31 are crucial in establishing the foundation for the formation of chicken spermatogonia stem cells and spermatogenesis. In this study, a series of recombinant vectors including varying lengths of the gga-miR-31 promoter were predicted and constructed. Through the utilization of the dual luciferase reporting system, the upstream −2180~0 bp region of gga-miR-31 was identified as its promoter region. Furthermore, it was predicted and confirmed that the activity of the gga-miR-31 promoter is increased by retinoic acid (RA). The binding of RA to the gga-miR-31 and Stra8 promoter regions was found to be competitive. Through the deletion of C-jun binding sites and the manipulation of C-jun expression levels, it was determined that C-jun inhibits the activity of the gga-miR-31 promoter. Furthermore, the combined treatment of C-jun and RA demonstrated that the positive regulatory effect of RA on the gga-miR-31 promoter is attenuated in the presence of high levels of C-jun. Overall, this study establishes a foundation for further investigation into the regulatory mechanisms of gga-miR-31 action, and provides a new avenue for inducing chicken embryonic stem cells (ESC) to differentiate into spermatogonial stem cells (SSC), and sperm formation.

## 1. Introduction

MicroRNA (miRNA) refers to a class of small non-coding RNAs approximately 22 nucleotides in length, which exert significant influence on organismal developmental processes including cell differentiation [1], spermatogenesis [2], and cancer formation [3]. Within the miRNA family, miR-31 actively participates in diverse regulatory mechanisms associated with development. Presently, the majority of investigations concerning miR-31 focus on its implications in disease pathology. Zhu et al. [4] demonstrated that miR-31-5p eliminated leukemia stem cells and inhibited the development of acute myeloid leukemia. Yu et al. [5] found that miR-31-5p contributed to tumor progression both in vitro and in vivo by directly targeting Special AT-Rich Sequence-Binding Protein 2 (SATB2)—it reversed the epithelial–mesenchymal transition and significantly increased activation of MEK/ERK signaling. Fong et al. [6] discovered that knockdown of miR-31 blocked esophageal squamous carcinogenesis in rats. However, the function of gga-miR-31 in poultry has rarely been studied; only our previous study found that gga-miR-31 plays a negative regulatory role in the process of meiosis in roosters, and its expression is decreased during the process of rooster germ cell differentiation.

The accurate transcription of miRNAs is governed by a multifaceted regulatory system, wherein the interactions between transcription factors and the promoter regions of miRNA precursors hold significant importance. The transcriptional regulation of miRNAs is influenced by a range of stimuli from the external environment and signals from different developmental stages. These stimuli prompt the binding of various transcription factors to transcriptional regulatory elements, resulting in the activation or repression of miRNA transcription. Consequently, the expression levels of mature miRNAs and their biological functions are affected, ultimately impacting the role of miRNAs in organism development. Yong et al. [7] found that myocardium-specific expression of miR-1 was regulated by the muscle tissue formation-related transcription factors SRF and MyoD. Yin et al. [8] found that in ovarian granulosa cells, SF-1 binds to the promoter region of miR-383 and promotes the expression of miR-383. Jiang et al. [9] found that the transcription factors SREBP and C/EBP regulate the promoter activity of miR-378. Although the transcriptional regulation of miR-31 has rarely been studied, Zhao et al. [10] demonstrated that HDAC3 decreased H3K9 acetylation in the promoter region of miR-31 in breast cancer, thereby inhibiting miR-31 transcription. Yang et al. [11] found that ALDH2 attenuated the inhibition of cardiotrophin levels by miR-31-5p, and its mutation or inhibition downregulated miR-31-5p levels. Despite this understanding, the specific factors that regulate the transcription of gga-miR-31, a crucial determinant of germ cell differentiation and meiosis formation in the cockerel, remain unexplored.

This study aims to predict and construct a series of gga-miR-31 promoter recombinant vectors with different length deletions, analyze their promoter activities, identify potential transcription factors that may bind to them, and combine retinoic acid (RA) induction and site-specific deletion techniques to determine the crucial factors influencing the transcriptional regulation of gga-miR-31-5p. The findings of this study provide a basis for further investigations into the functionality of gga-miR-31, and its potential applications in enhancing the efficacy of spermatogonial stem cells (SSC) formation and sperm production.

## 2. Results

### 2.1. gga-miR-31 Promoter Core Region Analysis

To further explore the role of gga-miR-31-5p in chicken germ cell differentiation and sperm formation, we analyzed the 2180 bp sequence upstream of gga-miR-31 (Figure 1A). Subsequently, three truncated fragments of varying lengths were constructed and transfected into DF-1 cells. The fluorescence activity of the samples was quantified using a dual luciferase reporter system after a 48 h collection period. Our findings indicate that the initiation activities of the Rugao yellow chicken gga-miR-31 promoter deletion fragment series had different initiation activities in DF-1 cells. Of the recombinant plasmids tested, pGL3-584 exhibited the highest activity, while both pGL3-1344 and pGL3-2180 showed significantly inhibited activities in comparison to pGL3-584. Additionally, there was a highly significant upregulation of pGL3-2180 activity in comparison to pGL3-1344. The results suggest the presence of significant negative regulatory elements within the −1344 bp~−584 bp region and significant positive regulatory elements within the −2180 bp~−1344 bp and −584 bp~+1 bp regions. Additionally, the results indicate that the −2180 bp~+1 bp region serves as the promoter region for gga-miR-31.

### 2.2. Bioinformatics Analysis of the Promoter Region of gga-miR-31

To further explore the transcriptional regulatory factors of gga-miR-31, a homology analysis was conducted on the upstream −2180~+1 bp region of miR-31 across various species (Appendix A) using MEGA7, and an evolutionary tree was constructed. Homology analysis revealed that the upstream promoter region of gga-miR-31 was 100% homologous to that of Zebra finch, but less homologous with other species. We then performed transcription factor prediction using AliBaba 2.1 (http://gene-regulation.com/pub/programs/alibaba2/index.html, accessed on 1 April 2020) for the −584 bp~+1 bp, −1344 bp~−584 bp and −2180 bp~−1344 bp regions, respectively (Appendix A, Figure 2B), and found that the initiation region of gga-miR-31 contains transcription factor binding sites for AP-1, ATF, and GR. We also found that the transcription factors Dl, CP1, ATF, p40x, Zen-1, HSF1_(long) only have binding sites in the −584 bp~+1 bp region, that C/EBP, IRF-1, ADR1, ATF-a, GCR1, C/EBP delta, GLO, Pit-1, and HOXA4 only have binding sites in the −1344 bp~−584 bp region, and that HNF-4 alpha1, C-jun, CFF, HSTF, NF-E2, ER, RAR-alpha1, C/EBP epsilon, 11-Oct, Pit-1a, HNF-1C, HNF-1, REB1, LyF-1, REV-Erb alpha, T3R-alpha, ETF, and PU.1 only have binding sites in the −2180 bp~−1344 bp region.

### 2.3. gga-miR-31 Is Transcriptionally Activated by RA(retinoic acid) and Competes with the Promoter of Stra8 to Bind RA

The miRNA-seq analysis of the RA-induced ESC to SSC-directed differentiation model established in the laboratory using high-throughput sequencing, revealed that gga-miR-31-5p expression was upregulated under this model (not available at this time). We also detected RAR-alpha transcriptional binding sites in the promoter region of gga-miR-31. We therefore speculate that the transcription of gga-miR-31 is most likely regulated by RA.

To further investigate the regulatory effect of RA on gga-miR-31-5p, we transfected each promoter fragment of gga-miR-31 and the promoter of Stra8 [12] (the promoter of Stra8 was used as a positive control for RA activation) into three cell types (DF-1, ESC (embryonic stem cells), and SSC (spermatogonial stem cells)), and stimulated each cell type after transfection, using RA. The results demonstrated that RA promoted both gga-miR-31 and Stra8 promoter activities in DF-1 cells, ESC, and SSC (Figure 3A). RA had the strongest effect on gga-miR-31 promoter activation in ESC, while gga-miR-31 promoter activity was significantly upregulated, though to a lesser extent than ESC, in DF-1 cells and SSC.

So, what is the relationship between RA, the gga-miR-31 promoter, and the Stra8 promoter? We used pEGFP-Bmp4 [13] as a negative control (pEGFP-Bmp4 is a pEGFP recombinant vector containing the promoter of lncRNA-Bmp4, which does not contain a RAR binding site in this promoter region and thus serves as a negative control for RA activation), replaced the 10^−5^ mol/L RA medium after transfecting DF-1 for 12 h, and collected cell samples for the dual luciferase activity reporter assay after 36 h. The results showed that compared with the cotransfected pEGFP-Bmp4 + pGL3-2180 group, the dual luciferase activity of the cotransfected pEGFP-Stra8 + pGL3-2180 group was significantly reduced. In addition, compared with the cotransfected pEGFP-Bmp4 + pGL3-Stra8 group, the dual luciferase activity of the cotransfected pEGFP-Stra8 + pGL3-2180 group was also significantly lower. This further suggests that the promoters of gga-miR-31 and Stra8 competitively bind RA (Figure 3B).

Interestingly, our findings indicate that the amount of RA in the ESC, PGC (primordial germ cell), and SSC cell types differed, with ESC < PGC < SSC (Figure 3C). Additionally, our previous study demonstrated that gga-miR-31-5p is down-regulated and Stra8 is upregulated in the ESC, PGC, and SSC cell types [14]. These results suggest that RA activates the gga-miR-31 promoter in different types of cells. However, it is important to note that there may be other factors influencing the regulation of gga-miR-31 transcription by RA, potentially due to differences between the cell types.

### 2.4. Effect of Transcription Factor C-jun Binding Sites on gga-miR-31 Promoter Activity

To further explore the factors that regulate gga-miR-31-5p expression during ESC to SSC differentiation, we conducted a screening of differential transcription factor binding sites in the promoters of gga-miR-31 and Stra8. Our analysis revealed the presence of 42 transcription factor binding sites that were exclusively found in the gga-miR-31 promoter region (Figure 4A). We analyzed the expression of these 42 transcription factors in ESC and SSC, in combination with the results of previous laboratory transcriptional sequencing experiments [15]. Among the differentially expressed transcription factors, AP-1, C-jun, HNF-4alp, HNF-1, HOXA4, ICSBP, and IRF1 were identified, with C-jun exhibiting the most significant difference in expression between ESC and SSC (Figure 4B).

To explore whether the transcription factor C-jun can bind to the gga-miR-31 promoter region and regulate its activity, we deleted the C-jun binding sites in the promoter region of gga-miR-31, and constructed deletion vectors: pGL3-2180-d1 (pGL3-2180-d1 means the deletion of the C-jun binding site 1 in PGL3-2180), pGL3-2180-d2 (pGL3-2180-d2 means the deletion of the C-jun binding site 2 in PGL3-2180), and pGL3-2180-d12 (pGL3-2180-d12 means the deletion of the C-jun binding sites 1 and 2 in PGL3-2180). Using pGL3-2180 as the positive control and PGL3-Basic as the negative control, cell samples were collected 48 h after transfection of DF-1 cells. The recombinant plasmids with deletions of the C-jun binding sites were detected by the dual luciferase reporter system. The results showed that compared with pGL3-2180, promoter activity was slightly upregulated when sites 1 and 2 were individually deleted, but the difference was not significant. The promoter activity was significantly upregulated when both sites 1 and 2 were deleted (Figure 4C).

### 2.5. Transcription Factor C-jun Inhibited the Transcription of gga-miR-31, and Combined with RA, Affected the Promoter Activity of gga-miR-31

In order to further explore the effect of C-jun on the promoter activity of gga-miR-31, the gga-miR-31 promoter recombinant plasmid pGL3-2180 was cotransfected into DF-1 cells with either the overexpression or interference vectors of C-jun. After 48 h, cell samples were collected for the dual luciferase activity assay. The results showed that compared with the pGL3-2180 + OENC control group, the activity of the pGL3-2180 group was significantly inhibited by overexpression of C-jun. Compared with the pGL3-2180 + KDNC control group, the activity of the pGL3-2180 group was significantly upregulated after interference with C-jun (Figure 5A).

After deletion of the C-jun binding sites, C-jun was overexpressed to further explore the influence of C-jun on the promoter activity of gga-miR-31. The overexpression vector of C-jun was cotransfected with pGL3-miR-31 deletion vectors into DF-1 cells. After 48 h, cell samples were collected and the activity of the samples was detected using the dual luciferase activity detection system. The results showed that compared with the pGL3-2180 + OE-C-jun group without deletion of the C-jun binding sites, the promoter activity was significantly enhanced after deletion of only one C-jun binding site and overexpression of C-jun, namely, the pGL3-2180-d1 + OE-C-jun group and the pGL3-2180-d2 + OE-C-jun group. The simultaneous deletion of two C-jun binding sites followed by overexpression of C-jun resulted in a highly significant enhancement of the initiation activity, compared with the above three groups (Figure 5B). All of these results indicate that the transcription factor C-jun affects the activity of the gga-miR-31 promoter, which is an important factor in regulating the expression of gga-miR-31-5p.

To explore the effects of C-jun and RA on the promoter activity of gga-miR-31, DF-1 cells were cotransfected with different vectors, with PRL-SV40 in each group. After 12 h, the 10^−5^ mol/L RA medium was replaced, and cell samples were collected after 36 h for dual luciferase activity detection assays. The results showed that, compared with the control group of pGL3-2180 + OENC, the activity of the pGL3-2180 promoter was significantly inhibited by overexpression of C-jun, and significantly enhanced by addition of RA. The results showed that, compared with the control group of pGL3-2180 + OENC, the activity of pGL3-2180 promoter was significantly inhibited by overexpression of C-jun, and significantly enhanced by the addition of RA. Simultaneous addition of RA and overexpression of C-jun significantly attenuated pGL3-2180 promoter activity, but it was higher than that in the group only overexpressing C-jun (Figure 5C). Therefore, RA has an enhancing effect on miR-31 promoter activity when the C-jun content in cells is low; when the C-jun content in cells is high, the inhibitory effect of C-jun on gga-miR-31 promoter initiation activity is stronger than the enhancing effect of RA on gga-miR-31 promoter initiation.

## 3. Discussion

In this study, we successfully identified the promoter region of gga-miR-31. Subsequently, we observed that RA (retinoic acid) exerts a positive regulatory effect on the transcription of gga-miR-31. Furthermore, we discovered that both the gga-miR-31 and Stra8 promoter regions competitively bind RA. C-jun was found to negatively regulate the transcription of gga-miR-31. Notably, the positive regulatory effect of RA on the gga-miR-31 promoter was attenuated in the presence of a high concentration of C-jun in cells (Figure 6). These findings serve as a fundamental basis for future utilization of gga-miR-31-5p to enhance the efficiency of SSC (spermatogonial stem cells) formation and sperm production.

In the realm of molecular regulation, akin to genes, transcriptional regulation exerts a significant influence on the expression of miRNAs, with the promoter assuming a crucial role at the transcriptional level. In this investigation, we scrutinized the −2180~0 bp upstream region of the gga-miR-31 miRNA precursor, revealing its possession of promoter activity.

The accurate transcription of miRNAs is governed by a multifaceted regulatory system, wherein the interplay between transcription factors and the promoter regions of miRNA constitutes a pivotal constituent. Regarding the transcription of miR-31, Ripamonti et al. [16] confirmed that BCL6 suppresses miR-31 expression by binding to its promoter in T cells. Tang et al. [17] found that in the mouse myogenic cell line C2C12, excessive addition of RA could promote the expression of miR-31-5p. In this study, an online prediction website was utilized to predict the presence of transcription factor binding sites, specifically those of RAR-alpha1, ATF, c/EBP, and HOXA4, within the promoter region of gga-miR-31. Additionally, it supported the idea that the activation of the gga-miR-31 promoter in chicken cells is induced by RA, aligning with the findings of our prior investigation, in which gga-miR-31-5p expression reached its peak on the fourth day of induction, in the RA-induced ESC differentiation to SSC model. Subsequently, the expression level gradually declined, yet remained higher than that of ESC (embryonic stem cells).

However, it was observed that the increase in activity of pGL3-2180 in SSC treated with RA was comparatively weaker than that in ESC. This prompts the question of whether there are additional factors, apart from RA, that regulate the promoter activity of gga-miR-31. Upon considering the sequencing results from Zhang et al. [15], it was discovered that C-jun, C/EBPalpha, GR, HNF-4a, HNF-1, HOXA4, ICSBP, IRF1, REV-ErbA, and RAP1 exhibited differential expression in ESC and SSC. Studies have shown that C-jun plays a potential regulatory role in the mitosis-to-meiosis transition [18], Rif1 regulates mESC pluripotency and self-renewal [19], and C/EBPalpha regulates neutrophil differentiation [20]. Further investigation is required to determine if these transcription factors play a crucial role in the differences in gga-miR-31-5p expression between ESC and SSC.

The intracellular proto-oncogene, transcription factor C-jun, can be rapidly and transiently expressed in response to gonadotropins, growth factors, phorbol esters, and neurotransmitters. Chieffi et al. [21] found that C-jun was highly expressed in spermatogonia, primary spermatocytes, secondary spermatocytes, and sperm cells. Xu et al. [22,23] demonstrated the antagonism of C-jun, and found that EDS-induced apoptosis of ex vivo Leydig cells was inhibited. C-jun promoted testosterone secretion in the basal state, promoted apoptosis of Leydig cells of the testis in rats and in human chorionic gonadotropin-induced rat testicular interstitial cells cultured in vitro.

Our previous research has demonstrated the inhibitory effect of gga-miR-31 on the process of sperm formation in roosters [14]. In this study, it was confirmed that the activity of the gga-miR-31 promoter is negatively regulated by C-jun, through various experiments involving deletion of C-jun binding sites, overexpression of C-jun and knockdown of C-jun. These findings provide further evidence of the significant role played by C-jun in rooster spermatogenesis, particularly through the gga-miR-31-5p-Stra8 axis.

Notably, it was observed that the combined overexpression of C-jun and increased RA content did not lead to an increase in the activity of pGL3-2180, suggesting that the inhibitory effect of C-jun on the gga-miR-31 promoter activity is stronger than the effect of RA. This is similar to Sp1 and Sp3, which, like C-Jun, play important roles in gene expression through the L-RARE (lamin A/C retinoic acid-responsive element) during RA treatment, as proposed by Okumura et al. [24]. However, further investigation is required to fully understand the association between C-jun and RA in chickens. Our findings establish a foundation for future research on the regulatory factors involved in the development of cock germ cells.

## 4. Materials and Methods

### 4.1. Ethics Statement

The fertilized eggs of Suqin yellow chickens used in this study were provided by the Institute of Poultry Science of the Chinese Academy of Agriculture Sciences. All the animal experimental procedures were approved and directed by the Yangzhou University Academic Committee, in accordance with the Jiangsu Province Experimental Animal Management Measures (License No. 45, Jiangsu Provincial Government, China) and the U.S. National Institutes of Health guidelines (NIH Pub. No. 85-23, revised 1996).

### 4.2. Materials

The fertilized eggs used in this experiment were provided by the Institute of Poultry Science of the Chinese Academy of Agriculture Sciences. DF-1 cells, pRL-SV40, pGL3.0-Basic, pEGFP-Bmp4 [13], pGL3-Stra8, pEGFP-Stra8 [12], OE-C-jun (OE-C-jun means overexpression vector of C-jun), OENC (OENC means negative control of C-jun overexpression vector), KD-C-jun (KD-C-jun means knockdown vector of C-jun), and KDNC (KDNC means negative control of C-jun knockdown vector) [25] are preserved in the laboratory.

### 4.3. Bioinformatics Analysis of Promoter

We obtain the 2180 bp gga-miR-31 (GeneID: NR_031499.1) upstream promoter sequences from the NCBI (National Center of Biotechnology Information) website (https://www.ncbi.nlm.nih.gov/, accessed on 1 February 2020). This region was analyzed by the online prediction programs BDGP: Neural Network Promoter Prediction (http://www.fruitfly.org/seq_tools/promoter.html, accessed on 1 February 2020), Promoter 2.0 Prediction Server (http://www.cbs.dtu.dk/servic-es/Promoter/, accessed on 1 February 2020), and TSSW (http://www.softberry.com/berry.phtml?topic=tssw&group=programs&subgroup=promoter, accessed on 1 February 2020). 

PROMO HOME PAGE (http://alggen.lsi. upc.es/cgi-bin/promo_v3/promo/prom oinit.cgi?dirDB=TF_8.3, accessed on 1 April 2020) and AliBaba 2.1 (http://gene-regulation.com/pub/programs/alibaba2/index.html, accessed on 1 April 2020) were comprehensively used to predict and analyze the potential transcription factor binding sites in the promoter region of gga-miR-31.

We obtained 2180 bp sequences upstream of miR-31 from NCBI for 16 species (Appendix A), including chicken, and used MEGA7 to analyze their homology and draw evolutionary trees.

### 4.4. Promoter Amplification and Vector Construction

According to the gga-miR-31 DNA sequence in NCBI, we cloned the 2180 bp upstream proximal end of the gga-miR-31 gene sequence. Based on the results of the promoter analysis, we constructed the plasmids pGL3-2180/+1, pGL3-1344/+1, and pGL3-584/+1 of gga-miR-31, containing the unidirectional deletions of the promoters. The primers were designed by NEBuilder (https://www.neb.com/, accessed on 1 February 2020) for amplification. 

To identify the C-jun binding sites on the chicken gga-miR-31 promoter, we conducted the site-deletion analysis via the deletion vector of the C-jun site that was constructed. We used the transcription factor binding site prediction results to design a C-jun binding site deletion vector (Figure 4C) and amplified each fragment, using whole genomic DNA extracted from 18.5 d Rugao yellow chicken testis tissue as the template. The site-deletion primers are listed in Appendix A.

Using XhoI and KpnI restriction sites, we subcloned the different plasmids containing the sequences of gga-miR-31 promoters into the pGl3-Basic vectors. We generated the luciferase reporter constructs, and we ligated the products using the 2× SoSoo Mix One Step Cloning Kit (Beijing Qingke Xinye Biotechnology Co., Ltd., Beijing, China).

Appendix A lists all primers used for the plasmid construction.

### 4.5. Cell Isolation and Culture

DF-1 is a chicken fibroblast cell line purchased from the ATCC library many years ago and kept in our lab. We cultured it using a high-sugar medium (Gibco, Grand Island, NY, USA) containing 10% FBS (fetal bovine serum) (10% FBS DMEM).

We isolated ESC from fresh fertilized eggs of the Rugao yellow chicken. The eggs were sterilized sequentially with benzalkonium bromide and 75% alcohol. The eggshell was cracked under aseptic conditions. The blastoderm in the egg yolk was separated with scissors and tweezers, washed 3 times in PBS (phosphate buffer saline), and then transferred to a 15 mL centrifuge tube. After centrifugation at 1000× *g* for 6 min, the upper layer of liquid was discarded. The cells were re-suspended using ESC special medium and cultured in a cell incubator. ESC with a good growth state were selected for follow-up experiments.

ESC special medium: DMEM (Sodium pyruvate, L-glutamine) + 10% FBS + 1% non-essential amino acid + 5.5 × 10^−5^ mol/L β-mercaptoethanol + 2% chicken serum + 10 ng/mL basic fibroblast growth factor + 0.1 ng/mL leukemia inhibitory factor + 5 ng/mL human stem cell factor + 100 mg/mL gentamicin sulfate.

We isolated SSC from male chicken embryos incubated at 37.5 °C and 60% humidity for 18.5 days. The eggs were sterilized with benzalkonium bromide and 75% alcohol in turn. Testicular tissue was taken in a sterile environment. The membranes and blood vessels on the testis were removed with tweezers. The remaining tissue was cut up with scissors, then 1 mg/mL type II collagenase was added and digested at 37 °C for 15–20 min. After centrifugation at 1000× *g* for 6 min, the upper layer of liquid was discarded. Then 0.25% trypsin was added for digestion, 10% FBS DMEM was added to terminate digestion. The cells were then filtered and collected using a 350-mesh filter cloth. After centrifugation at 1000× *g* for 6 min, the upper layer of liquid was discarded. The cells were re-suspended using SSC special medium and cultured at differential attachment for 45 min, repeated twice to remove the sertoli cells. SSC with a good growth state were selected for follow-up experiments.

SSC special medium: DMEM supplemented with 10% FBS + 2% chicken serum + 2 mmol/mL L-glutamine + 1 mmol/mL sodium pyruvate + 5.5 × 10^−5^ mol/mL b-mercaptoethanol + 5 ng/mL human stem cell factor + 10 ng/mL basic fibroblast growth factor + 0.1 ng/mL leukemia inhibitory factor + 10 ng/mL glial cell line-derived neurotrophic factor + 100 mg/mL gentamicin sulfate.

### 4.6. Transfection and Dual-Luciferase Assay

The DF-1, ESC, and SSC cell lines, in good growth conditions, were inoculated at 1 × 10^5^ cells per well into a 24-well plate, and when the cells grew to about 60% confluence, the pGL3 recombinant plasmid (500 ng) and the internal reference plasmid pRL-SV40 (14 ng) were cotransfected into the cells according to the Fugene (Promega, Fitchburg, WI, USA) instruction manual (https://fugene.com/wp-content/uploads/2021/07/FuGENE_HD-Quick-Protocol.pdf, accessed on 1 March 2020), using V_Fugene (μL)_:m_plasmid (μg)_ = 3:1. While a negative control group (pGL3-Basic plasmid cotransfected with pRL-SV40 plasmid) was set up, with repeats in 3 wells. After the 48 h transfection, the cells were lysed and collected. We analyzed the relative luciferase activities using the Dual-Luciferase Reporter Assay System (Promega, Fitchburg, WI, USA), according to the method of Wang et al. [14].

The DF-1 cells in a good growth condition were inoculated at 1 × 10^5^ cells per well into a 24-well plate. When the cells grew to about 60% confluence, transfection was performed according to the Fugene instruction manual. The following groups were set up: ➀ pGL3-Basic (500 ng) + OENC (500 ng) + pRL-SV40 (14 ng); ➁ pGL3-2180 (500 ng) + OENC (500 ng) + pRL-SV40 (14 ng); ➂ pGL3-2180 (500 ng) + OE-C-jun 500 ng) + pRL-SV40 (14 ng); ➃ pGL3-2180 (500 ng) + OENC (500 ng) + pRL-SV40 (14 ng) + RA; ➄ pGL3-2180 (500 ng) + OE-C-jun (500 ng) + pRL-SV40 (14 ng) + RA. After the 12 h transfection, the medium was replaced; groups ➀, ➁, and ➂ were cultured using 10% FBS DMEM, and groups ➃ and ➄ with 10^−5^ mol/L RA, 10% FBS DMEM, and cells were harvested 36 h later for dual luciferase activity assays.

The transfection conditions involving the cotransfection of the pGL3 recombinant plasmid with the vector of C-jun, were 500 ng of the pGL3 recombinant plasmid, 14 ng of pRL-SV40, 500 ng of the OENC/OE-C-jun/KDNC/KD-C-jun plasmid, and 3.04 μL of the Fugene transfection reagent.

### 4.7. ELISA for RA Concentration

Chicken ESC, PGC, and SSC were isolated separately according to the method of Zhang [15] and counted by a live cell counter. Each type was counted in groups of 1 × 10^6^ and resuspended by adding 150 μL PBS. Cells were lysed by ultrasonic lysis in an ice bath at 30% power, set at crushing for 5 s and intermittent for 5 s, for a total of 5 min. The supernatant was taken after centrifugation at 2000 rpm/min at 4 °C for 20 min. The RA concentration was measured using the RA ELISA kit (Shanghai Danshi Biotechnology Co., Ltd., Shanghai, China). Standard wells and sample wells were set, then 50 μL of standards of different concentrations were added to each of the standard wells, 50 μL of the sample was added to the sample wells, and 100μL enzyme conjugate reagent was added to each well. Wells were incubated at 37 °C for 1 h. The liquid was then discarded, and the plate shaken dry. Each well was filled with washing solution, let to stand for 30 s and then the liquid was discarded; this was repeated 5 times and then the plate was patted dry. Color-developing agent A, 50 μL, and then color-developing agent B, 50 μL, were added to each well. The mixture was mixed gently by shaking, and the color-developing process was carried out at 37 °C for 15 min. Termination solution, 50 μL, was added to each well to terminate the reactions. Absorbance was measured at 450 nm. According to the OD value of the sample, the actual concentration was calculated using a standard curve.

### 4.8. Data Analysis

Difference analysis was performed using SPSS 19.0. *p* < 0.05 was considered to have a significant difference. The histograms were drawn by GraphPad Prism 6.

### 4.9. Data Availability

All the data required by the author to confirm the conclusions reached in the article are fully reflected in the article.

## 5. Conclusions

This study revealed that the 2180 bp region upstream of gga-miR-31 is its promoter region, and that gga-miR-31 transcription is positively regulated by RA and competes with Stra8 to bind RA. In addition, the transcription factor C-jun was found to act as a transcriptional repressor to inhibit gga-miR-31 transcription and to attenuate the positive regulation of the gga-miR-31 promoter by RA, when large amounts of C-jun were present in the cells.

## Figures and Tables

**Figure 1 ijms-24-12516-f001:**
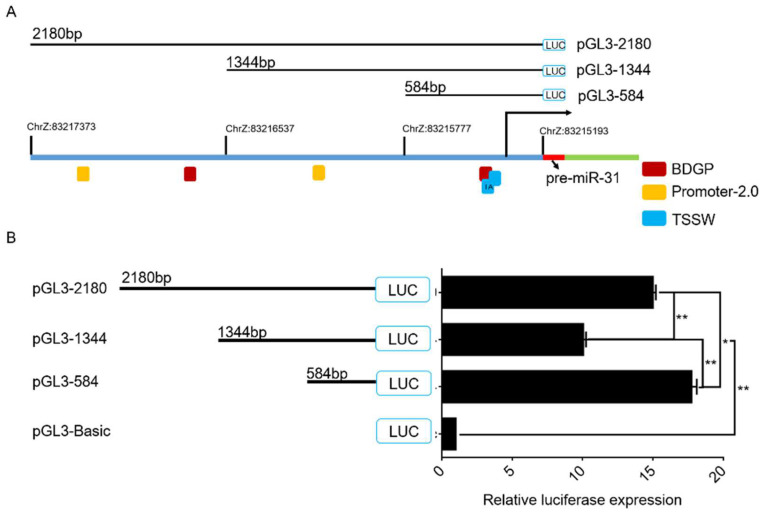
gga-miR-31 promoter core region analysis. (**A**) Online prediction of gga-miR-31 promoter region. Quadrilaterals filled with red, yellow, and light blue in Figure 1 mean we used online prediction software to predict the promoter sites contained in 2180 bp upstream of gga-miR-31; BDGP, promoter 2.0, and TSSW, respectively. (**B**) Dual luciferase activity assay with different truncated carriers for relative luciferase activity. Note: For PGL3-2180, the target fragment (−2180~+1 bp, chrZ:83217373-83215193) was ligated to the pGL3-Basic vector. For PGL3-1344, the target fragment (−1344~+1 bp, chrZ:83216537-83215193) was ligated to the pGL3-Basic vector. For PGL3-584, the target fragment (−584~+1 bp, chrZ:83215777-83215193) was ligated to the pGL3-Basic vector. The marker “*” represents a significant difference (*p* < 0.05), while “**” represents a highly significant difference (*p* < 0.01).

**Figure 2 ijms-24-12516-f002:**
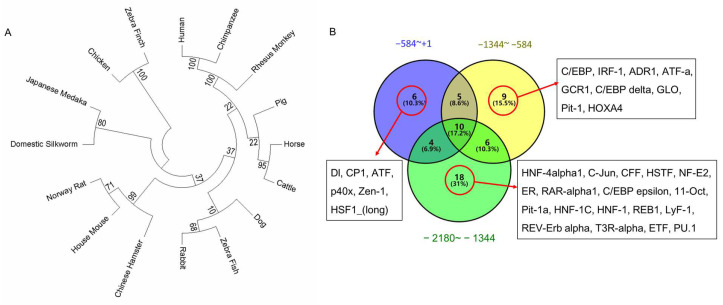
Conservation of the promoter region and the prediction of binding transcription factors. (**A**) gga-miR-31 promoter sequence (−2180~+1 bp) phylogenetic tree of sixteen species; (**B**) Predicted transcription factors that may bind to the promoter region of gga-miR-31.

**Figure 3 ijms-24-12516-f003:**
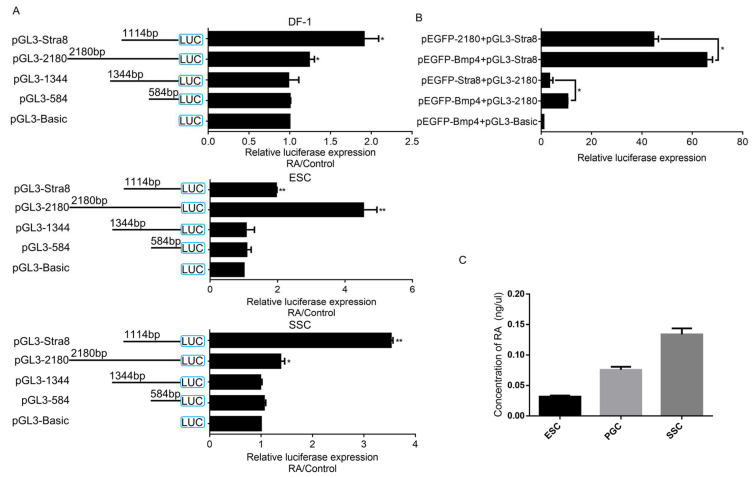
Activation of gga-miR-31 transcription by RA. (**A**) The DF-1, ESC, and SSC cell types were transfected with pGL3-miR-31 and pGL3-Stra8. Twelve hours after transfection, the fluid was changed and one group was cultured with 10^−5^ mol/L RA, 10% FBS DMEM, and the other group was cultured with 10% FBS DMEM. After 36 h, the cells were collected for dual luciferase activity assays and calibrated with the control group. Relative activity was calculated as RA group/Control group. (**B**) Using pEGFP-Bmp4 as a negative control for the RAR binding site, different groups were set up to cotransfect DF-1 cells with the gga-miR-31 promoter and the Stra8 promoter for 12 h. The dual luciferase activity was detected after treatment with 10^−5^ mol/L RA, 10% FBS DMEM. (**C**) Determination of RA concentration in different cell types. The marker “*” represents a significant difference (*p* < 0.05), while “**” represents a highly significant difference (*p* < 0.01).

**Figure 4 ijms-24-12516-f004:**
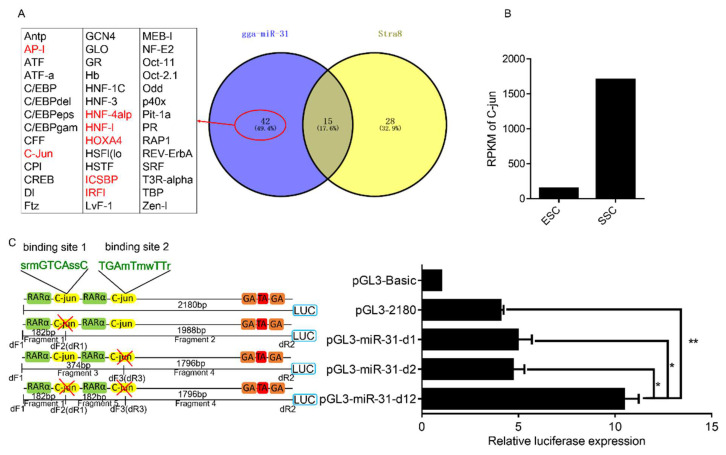
Effect of deletion of the C-jun binding site on the activity of the gga-miR-31 promoter. (**A**) Venn analysis of the transcription factor binding sites of the gga-miR-31 and the Stra8 promoter regions. The transcription factors marked in red are the transcription factors differentially expressed between ESC and SSC. (**B**) RNA-seq data of C-jun in ESC and SSC. (**C**) Relative fluorescence activity of various recombinant plasmids after deletion of C-jun binding site in the promoter region of gga-miR-31. The marker “*” represents a significant difference (*p* < 0.05), while “**” represents a highly significant difference (*p* < 0.01).

**Figure 5 ijms-24-12516-f005:**
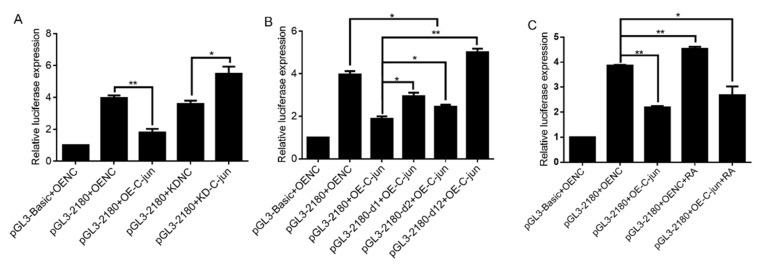
The transcription factor C-jun inhibited the transcription of gga-miR-31, and combined with RA, affected the promoter activity of gga-miR-31. (**A**) Effect of overexpression/knockdown of C-jun on pGL3-2180 vector activity. (**B**) Effect of overexpression of C-jun on gga-miR-31 promoter activity after deletion of the C-jun binding sites. (**C**) Effect of C-jun in combination with RA on gga-miR-31 promoter activity. OE-C-jun means the overexpression vector of C-jun. OENC means negative control of the C-jun overexpression vector. KD-C-jun means the knockdown vector of C-jun. KDNC means negative control of the C-jun knockdown vector. The marker “*” represents a significant difference (*p* < 0.05), while “**” represents a highly significant difference (*p* < 0.01).

**Figure 6 ijms-24-12516-f006:**
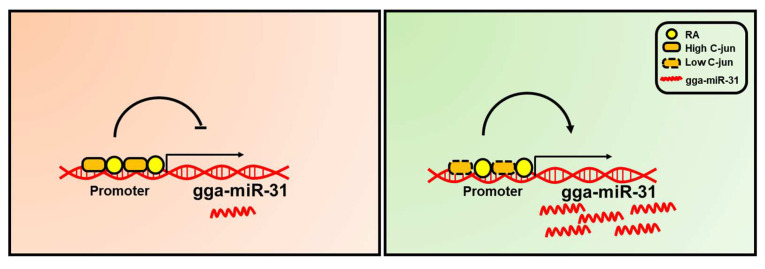
Schematic drawing of how RA and C-jun regulate the transcription of gga-miR-31 in chicken cells.

## Data Availability

The original data in the article can be obtained directly from the first author. The data are not publicly available due to privacy restrictions.

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
