# Peer review of "Analysis of the Promoter Regions of gga-miR-31 and Its Regulation by RA and C-jun in Chicken"

_ijms, 2023, doi:10.3390/ijms241512516_

Round 1

Reviewer 1 Report

-

Author Response

Dear editor and reviewer:

I have revised the manuscript according to the review comments, and answered each question one by one, the red color is the corresponding answer for each question.

Reviewer 2 Report

Reviewer’s comment to authors:

Wang and colleagues identified that the 2180bp region upstream of gga-miR-31 is its promoter region, and gga-miR-31 transcription is positively regulated by RA and competes with Stra8 to bind RA in DF-1/ESC/SSC. The authors also found the transcription factor C-jun act as a transcriptional repressor to inhibit gga-miR-31 transcription and to attenuate the positive regulation of the gga-miR-31 promoter by RA when large amounts of C-jun were present in the cells.

This manuscript is very interesting for this reviewer however, to accept this manuscript in this journal, please answer all question, add a little bit detail information and revise some figures. Before accepting this manuscript, this reviewer needs to review again.   

This reviewer did not check for English errors, please ask somebody professional to check English used in this entire document, figures and figure legends before submission.   

Page 1;

When authors cite some papers in the text, only first author’s name was cited, however this reviewer respect all authors in the papers because in many cases, the work was accomplished as a team not solo.

This reviewer recommends to add “et., al.” after first author’s name, but please follow this journal’s regulation.

Examples:    

Page1.

Line 33 “Zhu” should be “Zhu et al.,”

Line 34: “Yu” should be “Yu et al.,”

Line 37: Fong et al.,

Line 39: “gga-miR-31” what does “gga” abbreviate for?

Page2.

Line 49 “Yong” should be “Yong et al.,” .

Line 51: “Yin[8]” should be “Yin[8] et a;.,” .

Page3:

Figure 1

What do they mean by quadrilaterals filled with red, yellow, and light blue in figure 1?

No explanation in figure legend and main test.

Figure2

what does the percentage means in B? No explanation in figure legend and main test. 

Page 8: line 295: “DF-1 cells”

What is the cell (cell line?)? Where did the authors purchase this? What is the resource of this cells?

Page 4:

ESC, SSC etc.  What do these abbreviations mean? The authors should use abbreviation carefully and when the authors use these abbreviations in the text, they should add full name of these abbreviations describe in the section of “Material and Method” or add list as the supplement table.

Fig 3

The letters in the figures are very small and it is very hard to see them.

So, the authors should use bigger fonts and remake this Figure3.

Page 5

Line 159: “SSC[16] We” should be “SSC[16]. We”.

what does the percentage means in A? No explanation in figure legend and main test. 

The letters in the figures are very small and it is very hard to see them.

Page6

What is “L-RARE”?

Page 9

Line326-

You need to cite the information of for these basic vectors such as pGL3.0-Basic, from where authors purchased such as the company name etc.

How did the authors make overexpression C-jun and RA contract?

Line 331: “DF-1/ESC/SSC”

There are no detailed information how the authors isolate these cells not only in this text but also not in reference “Zhang [16]”. Please write more details. After isolation how did authors maintain the cells before use for experiment etc?  what media did authors use?    

Line 332 “1×105 cells”?

Line 335: “Fugene instruction manual” what is this? Manual from the company? If so, need reference or web site information here.

What methods authors use for transfection? Authors needs more information for this method.

Line 343 “1×106” ?

Author Response

Dear editor and reviewer:

I have revised the manuscript according to the review comments, and answered each question one by one, the red color is the corresponding answer for each question.

Reviewer2:

  1. Page 1;When authors cite some papers in the text, only first author’s name was cited, however this reviewer respect all authors in the papers because in many cases, the work was accomplished as a team not solo. This reviewer recommends to add “et., al.” after first author’s name, but please follow this journal’s regulation. Examples:   Page1.: Line 33 “Zhu” should be “Zhu et al.,” Line 34: “Yu” should be “Yu et al.,” Line 37: Fong et al. Page2. : Line 49 “Yong” should be “Yong et al.,” . Line 51: “Yin[8]” should be “Yin[8] et a;.,” .

Thanks for your advice. I have revised all the references in the article to the first author et., al. and marked them in red.

  1. Page1,Line 39: “gga-miR-31” what does “gga” abbreviate for?

Thanks for your advice. The abbreviation “gga” in gga-miR-31 is abbreviation of gallus gallus, the roman name for chicken.

  1. Page3: ,Figure 1:What do they mean by quadrilaterals filled with red, yellow, and light blue in figure 1?No explanation in figure legend and main test.

Thanks for your advice. Quadrilaterals filled with red, yellow, and light blue in figure 1 mean we used online prediction software BDGP, promoter 2.0 and TSSW to predict the promoter sites contained in 2180bp upstream of gga-miR-31, respectively. We’ve added it to the figure legend of figure1 and identified it in red.

  1. Page3: Figure2

what does the percentage means in B? No explanation in figure legend and main test. 

 Thanks for your advice. The percentage in Figure2B, refers to the proportion of the number of transcription factors in each region in the total number of transcription factors types in the gga-miR-31 promoter region. For example, the number of transcription factors only have binding sites in the -584bp~+1bp region is 6. It accounts for 10.3% of the transcription factor types in the-2180bp~ + 1bp region of gga-miR-31 promoter.

  1. Page 8: line 295: “DF-1 cells”. What is the cell (cell line?)? Where did the authors purchase this? What is the resource of this cells?

Thanks for your advice. DF-1 is a chicken fibroblast cell line purchased from the ATCC library many years ago and kept in our lab.

  1. Page 4: ESC, SSC etc.  What do these abbreviations mean? The authors should use abbreviation carefully and when the authors use these abbreviations in the text, they should add full name of these abbreviations describe in the section of “Material and Method” or add list as the supplement table.

Thanks for your advice. ESC means embryonic stem cell. SSC means spermatogonial stem cell. The full name of abbreviations in this article have been added when they first showed up and marked them in red.

  1. Fig 3. The letters in the figures are very small and it is very hard to see them.

So, the authors should use bigger fonts and remake this Figure3.

Thanks for your advice. We have changed the font size in figure3.

8.Page 5 Line 159: “SSC[16] We” should be “SSC[16]. We”.

Thanks for your advice. We have modified it and marked it in red.

9.what does the percentage means in A? No explanation in figure legend and main test. The letters in the figures are very small and it is very hard to see them.

 Thanks for your advice. The percentage in Figure4A, refers to the proportion of the number of transcription factors in each region in the total number of transcription factors types in the gga-miR-31 promoter and Stra8 promoter. For example, the number of transcription factors only have binding sites in gga-miR-31 promoter is 42. It accounts for 49.4% of the transcription factor types in the gga-miR-31 promoter and Stra8 promoter.

We have changed the font size in figure4.

10.Page6 What is “L-RARE”?

Thanks for your advice. L-RARE means lamin A/C retinoic acid-responsive element. I have added its full name to the article and marked it in red.

  1. Page 9 Line326-

You need to cite the information of for these basic vectors such as pGL3.0-Basic, from where authors purchased such as the company name etc.

Thanks for your advice. pGL3.0-Basic was purchased from Honorgene and is kept by our lab.

  1. How did the authors make overexpression C-jun and RA contract?

Thanks for your advice. The DF-1 in good growth condition were inoculated with 1×105 cells per well in a 24-well plate. When the cells grew to about 60%, transfection was performed according to the Fugene instruction manual. The following groups are set up: â‘  pGL 3-Basic (500 ng) + OENC (500 ng) + pRL-SV40 (14 ng); â‘¡pGL3-2180(500ng)+OENC(500ng)+pRL-SV40(14 ng); â‘¢pGL3-2180(500ng)+ OE-C-jun 500ng)+pRL-SV40(14 ng); â‘£pGL3-2180(500ng)+ OENC(500ng) +pRL-SV40(14 ng)+RA; ⑤pGL3-2180(500ng)+ OE-C-jun(500ng)+pRL-SV40(14 ng)+RA. Transfection for 12h, the medium was replaced, where groupsâ‘ â‘¡â‘¢ were cultured using 10% FBS`DMEM and groups④⑤ with 10-5 mol/L RA`10% FBS`DMEM, and cells were harvested 36 h later for dual luciferase activity. We have added this section to Materials and Methods 4.6 and marked it in red.

13.Line 331: “DF-1/ESC/SSC”

There are no detailed information how the authors isolate these cells not only in this text but also not in reference “Zhang [16]”. Please write more details. After isolation how did authors maintain the cells before use for experiment etc?  what media did authors use?    

Thanks for your advice. I am very sorry that I should have cited the author's doctoral thesis when I inserted the reference and these three cell isolation methods were described in detail in his graduation thesis. At present, I have revised the cited references. The ESC/SSC isolation method and the medium components of the three cells were added to the 4.5 Cell isolation and culture in the Materials and Methods.

DF-1 is a chicken fibroblast cell line purchased from the ATCC library many years ago and kept in our lab. We cultured it using a high-sugar medium (Gibico) containing 10% serum (10% FBS`DMEM).

We isolated ESC from fresh fertilized eggs of Rugao yellow chicken. The eggs were sterilized sequentially with benzalkonium bromide and 75% alcohol. The eggshell was cracked under aseptic environment. The blastoderm in the egg yolk was separated with scissors and tweezers, washed 3 times in PBS, and then transferred to a 15ml centrifuge tube. After centrifugation at 1000 g for 6 min, the upper layer of liquid was discarded. The cells were re-suspended using ESC special medium and cultured in a cell incubator. ESC with good growth state was selected for follow-up experiment.

ESC special medium : DMEM (Sodium pyruvate, L-glutamine) +10% FBS + 1% non-essential amino acid +5.5 × 10-5 mol/L b-mercaptoethanol + 2% chicken serum + 10 ng/mL basic fibroblast growth factor +0.1 ng/mL leukemia inhibitory factor + 5 ng/mL human stem cells factor +100mg/mL gentamicin sulfate.

We isolated SSC from male chicken embryos incubated at 37.5℃ and 60 humidity for 18.5 days. The eggs were sterilized with benzalkonium bromide and 75% alcohol in turn. Testicular tissue was taken in a sterile environment. The membranes and blood vessels on the testis were removed with tweezers.The remaining tissue was cut up with scissors, then 1 mg/mL type II collagenase was added and digested at 37℃ for 15-20 minutes. After centrifugation at 1000g for 6 min, the upper layer of liquid was discarded. Then 0.25% trypsin was added for digestion, 10% FBS 'DMEM was added to terminate digestion. The cells were then filtered and collected using a 350-mesh filter cloth. After centrifugation at 1000g for 6 min, the upper layer of liquid was discarded. The cells were re-suspended using SSC special medium and cultured at differential attachment for 45 minutes, repeated twice to remove the adherent cells. SSC with good growth state was selected for follow-up experiment.

SSC special medium: DMEM supplemented with 10% FBS+2% chicken serum+2 mmol/mL L-glutamine+ 1 mmol/mL sodium pyruvate+5.5 ×10-5mol/mL b-mercaptoethanol +5 ng/mL human stem cells factor+10 ng/mL basic fibroblast growth factor +0.1 ng/mL leukemia inhibitory factor +10 ng/mL glialcellline-derived neurotrophic factor +100mg/mL gentamicin sulfate.

  1. Line 332 “1×105 cells”?

Thanks for your advice. Here is the format error. 1×105 cells mean 1×105 cells,I have modified it and marked it in red.

  1. Line 335: “Fugene instruction manual” what is this? Manual from the company? If so, need reference or web site information here.

What methods authors use for transfection? Authors needs more information for this method.

Thanks for your advice. Transfection reagent Fugene was purchased from promaga and we refer to its instructions for using from this website https://fugene.com/wp-content/uploads/2021/07/FuGENE_HD-Quick-Protocol.pdf. We incubated VFugene (μL): mplasmid (μg) = 3:1 at room temperature for 15min, and added it to the culture plate with the new medium for transfection. The plasmid here is the pGL3 recombinant plasmid (500 ng) and the internal reference plasmid pRL-SV 40 (14 ng).

16.Line 343 “1×106” ?

Thanks for your advice. Here is the format error. 1×106 cells mean 1×106 cells,I have modified it and marked it in red.

Reviewer 3 Report

The study is valuable to deeply understand the mechanisms of germ cells development. The authors used start of the art methodology to provide clear evidence for the possible regulation of c-jun in the development of spermatogonial cells. 

My comments:

1. Write complete names of the abbreviations.

2. It is important to show correlation between j-jun and proliferation/differentiation (immunostaining with pre-meiotic and meiotic cell markers) of the germ cells used. This in parallel to the effect of RA.

Moderate

Author Response

Dear editor and reviewer:

I have revised the manuscript according to the review comments, and answered each question one by one, the red color is the corresponding answer for each question.

Reviewer 3:

My comments:

  1. Write complete names of the abbreviations.

Thanks for your advice. The full name of abbreviations in this article have been added when they first showed up and marked them in red.

  1. It is important to show correlation between c-jun and proliferation/differentiation (immunostaining with pre-meiotic and meiotic cell markers) of the germ cells used. This in parallel to the effect of RA.

Thanks for your advice. This part of the experiment is downstream of our ongoing experiments, where we are currently exploring the specific function of c-jun during germ cell differentiation.

Round 2

Reviewer 2 Report

Reviewer’s comment to authors:

Wang and colleagues identified that the 2180bp region upstream of gga-miR-31 is its promoter region, and gga-miR-31 transcription is positively regulated by RA and competes with Stra8 to bind RA in DF-1/ESC/SSC. The authors also found the transcription factor C-jun act as a transcriptional repressor to inhibit gga-miR-31 transcription and to attenuate the positive regulation of the gga-miR-31 promoter by RA when large amounts of C-jun were present in the cells.

This manuscript is very interesting for this reviewer. This is a re-revised manuscript. The authors corrected the original manuscript following the reviewers’ comments as much as they could.

However, this reviewer did not check for English errors. I suggest editor to ask somebody professional to check English used in this entire document, figures and figure legends before submission.   

Then, please accept this manuscript.

I recommend to the editors to accept it.

Author Response

Dear editor and reviewer:

Thank you very much for your valuable comments on this article. I have gotten a professional to revise the language of the article and marked the changes in red.

Reviewer 3 Report

The reviewer suggest to provide some preliminary results (immunostaining) that show/support the correlation between c-jun and proliferation/differentiation of the germ cells used. This in parallel to the effect of RA. 

Author Response

Dear editor and reviewer:

I have revised the manuscript according to the review comments, and answered each question one by one, the red color is the corresponding answer for each question.

Reviewer 3:

The reviewer suggest to provide some preliminary results (immunostaining) that show/support the correlation between c-jun and proliferation/differentiation of the germ cells used. This in parallel to the effect of RA.

Thanks for your advice. We are sorry that we are unable to provide some results on the correlation between C-jun and germ cell proliferation/differentiation for the time being. Because SSC need to be isolated from the testis of male chicken embryos incubated for 18.5 days and the time required to carry out the relevant experiments is long, we are unable to provide the relevant results before the revision date. However, our previous study demonstrated that MAPK8-mediated JNK signaling promotes the differentiation of ESC to male germ cells, along with the up-regulated expression of C-jun (Jun) during this process(MAPK8 regulates chicken male germ cell differentiation through JNK signaling pathway, Figure2B).

Our unpublished experimental results also demonstrated that an activator of the JNK signaling pathway (20 umol/L Anisomycin) upregulated C-jun expression and increased its promoter activity, while its inhibitor (10 umol/L SP600125) also inhibited C-jun promoter activity. In conclusion, it can be hypothesized that there is a relationship between C-jun and germ cell differentiation.

We will conduct experiments related to the function of C-jun, and hope to find out the role of C-jun in the process of germ cell proliferation/differentiation as soon as possible, and we look forward to your continued valuable suggestions.

(MAPK8 regulates chicken male germ cell differentiation through JNK signaling pathway).

FIGURE 2 In vitro inhibition of JNK signaling pathway impaired the formation of PGCs cells. (A) Cell morphology evaluated by indirect immunofluorescence assay in different cell groups(scale bar: 20 µm). (B) Changes of the reproductive marker genes expression and MAPK8 downstream genes expression in different groups. (C) Production of PGCs in different groups according evaluated by fluorescence activated cell sorting. (*P < 0.05, **P < 0.01, Bars represent SD, n = 3. The statistical significance was derived by comparing each group with the control group)
